# Effect of Penetration Enhancers on Toenail Delivery of Efinaconazole from Hydroalcoholic Preparations

**DOI:** 10.3390/molecules26061650

**Published:** 2021-03-16

**Authors:** Jun Soo Park, Jeong Soo Kim, Myoung Jin Ho, Dong Woo Park, Eun A. Kim, Yong Seok Choi, Sun Woo Jang, Myung Joo Kang

**Affiliations:** 1Colledge of Pharmacy, Dankook University, Dongnam-gu, Cheonan 330-714, Chungnam, Korea; chon523273@gmail.com (J.S.P.); butable@gmail.com (M.J.H.); eastkingdw@hanmail.net (D.W.P.); anemone014@naver.com (E.A.K.); analysc@dankook.ac.kr (Y.S.C.); 2Dong-A Pharmaceutical Co. Ltd., Giheung-gu, Yongin 446-905, Gyeonggi, Korea; js_kim@donga.co.kr (J.S.K.); chb@donga.co.kr (S.W.J.)

**Keywords:** efinaconazole, transungual drug delivery, hydroalcoholic solution, permeation enhancer, bovine hoof, onychomycosis, propylene glycol dicaprylocaprate, hydroxypropyl-β-cyclodextrin

## Abstract

The incorporation of permeation enhancers in topical preparations has been recognized as a simple and valuable approach to improve the penetration of antifungal agents into toenails. In this study, to improve the toenail delivery of efinaconazole (EFN), a triazole derivative for onychomycosis treatment, topical solutions containing different penetration enhancers were designed, and the permeation profiles were evaluated using bovine hoof models. In an in vitro permeation study in a Franz diffusion cell, hydroalcoholic solutions (HSs) containing lipophilic enhancers, particularly prepared with propylene glycol dicaprylocaprate (Labrafac PG), had 41% higher penetration than the HS base. Moreover, the combination of hydroxypropyl-β-cyclodextrin with Labrafac PG further facilitated the penetration of EFN across the hoof membrane. In addition, this novel topical solution prepared with both lipophilic and hydrophilic enhancers was physicochemically stable, with no drug degradation under ambient conditions (25 °C, for 10 months). Therefore, this HS system can be a promising tool for enhancing the toenail permeability and therapeutic efficacy of EFN.

## 1. Introduction

Onychomycosis is one of the most chronic and prevalent nail disorders, caused by fungal infection. Epidemiological studies suggest about 19% of the global population is affected by onychomycosis caused by dermatophytes (*Trichophyton rubrum* and *Trichophyton mentagrophytes*), non-dermatophyte molds, and yeasts (*Candida albicans*) [1,2]. Efinaconazole (EFN, Figure 1), the first topical triazole derivative, is prescribed for the treatment of toenail onychomycosis. This antifungal agent has been demonstrated to be potent against *Trichophyton rubrum*, *Trichophyton mentagrophytes*, and *Candida albicans* in toenail by obstructing ergosterol biosynthesis, presumably through sterol 14-demethylase inhibition [3,4]. Topical application of EFN solution (10% w/w) was markedly efficacious than other topical treatments with amorolfine or ciclopirox against mild to moderate subungual onychomycosis [5]. Currently, the marketed product of EFN is designed as a hydroalcoholic topical solution (Jublia, Kaken Pharmaceutical Co., Ltd. (Tokyo, Japan)) [6]; ethanol is employed as a volatile vehicle to dissolve the antifungal agent. In addition, to enhance the skin and/or nail delivery of EFN, lipophilic enhancers, such as myristyl lactate and diisopropyl adipate, are included, along with cyclomethicone as a wetting agent. Upon topical application, the organic vehicle is rapidly vaporized, providing high drug concentration onto the nail plate. Subsequently, EFN molecules diffuse through the nail plate with lipophilic permeation enhancers. 

In designing topical preparations of antifungal agents, especially for toenail treatment, endowing satisfactory penetration into the relevant layer, including the nail plate, is one of the pivotal factors influencing therapeutic efficacy [7,8,9]. The nail plate is a thin (0.25–1.00 mm), hard, and convex-shaped structure, which is principally made of fibrous proteins, approximately 80–90 layers of keratinized cells, and keratin filaments [10,11]. These components are tightly junctioned via a disulfide link which is responsible for the toughness and barrier function of nails. Most antifungal agents are lipophilic and possess high affinity for keratinized tissues, including the toenail, which can have a deleterious effect on drug absorption, impeding drug penetration to deeper layers of the nail plate [12]. Although EFN has a relatively lower binding affinity to keratinized tissue compared to several lipophilic antifungal agents [5], topical preparation possessing an enhanced penetration profile is still required to improve the therapeutic efficacy of the triazole derivative in patients with onychomycosis. Actually, the increased topical delivery of EFN markedly accelerated the structural recovery of keratin layer in a guinea pig onychomycosis model [13].

To enhance the ungual delivery of antifungal agents, different strategies have been utilized, such as chemical modification of antifungal agents, employment of penetration enhancers, nanocarrier-based delivery, and mechanical or physical approaches, including iontophoresis, photodynamic therapy, and lasers [11,14,15]. The incorporation of permeation enhancers in topical preparations, such as lacquer, cream, or hydrogel, have been widely explored to increase drug penetration into the toenail, as a simple and effective technique [8,13]. Saner et al. (2014) reported that topical treatment with lipophilic excipients principally altered the microstructure of the nail plate through physical interaction with the lipid components of the plate, promoting drug penetration [16]. On the other hand, hydrophilic enhancers, such as polyethylene glycols, pantothenic acid, hydroxypropyl-β-cyclodextrin (HP-β-CD) or sodium lauryl sulfate (SLS) has been shown to increase drug penetration across the nail plate by improving the hydration level of the nail plate, forming a porous microchannel, or by decreasing the contact angle between topical solution and the surface or nail plaque [17,18,19,20]. As the aforementioned chemical approach relies on the chemical structure and physicochemical nature of therapeutic agents, a case-specific formulation study is essential to find the finest composition.

The objective of the present study was to design a topical preparation for EFN with high ungual permeation profile. Different hydroalcoholic solutions (HSs) containing several lipophilic enhancers (L-HSs, Table 1) were formulated, and their physicochemical properties and permeation profile through the bovine hooves were evaluated. After selecting a lipophilic enhancer, HSs containing both lipophilic and hydrophilic permeation enhancers (LH-HSs, Table 2) were further designed, and their permeation profiles were evaluated with L-HSs including formula similar to the marketed product (L-HS1), or several oil solutions (OSs, Table 3). The physicochemical stability of the optimized topical formula was assessed under stress and ambient conditions.

## 2. Results

### 2.1. Effect of Lipophilic Penetration Enhancer on Transungual EFN Delivery

In the present study, topical preparations containing different penetration enhancers were formulated to facilitate toenail delivery of EFN, and permeation profiles through the bovine hoof slice were evaluated in a Franz diffusion cell model. First, ethanol-based HS formulations containing different L-HSs were assessed for EFN delivery (Table 1). Volatile vehicles account for more than 50% of the total weight of preparations, not only to provide high drug loading (10% *w*/*w*) with excellent solubilizing capacity but also to promote drug diffusion and/or penetration into nail plate by increasing concentration gradient after vaporization. As lipophilic absorption enhancers, several water-immiscible oily excipients, such as Lauroglycol 90, Labrafac PG, medium-chain triglyceride, isopropyl myristate, and isopropyl palmitate, were employed. Saner et al. (2014) reported that these lipophilic ingredients altered the microstructure of the nail plate by interacting with its lipid components, promoting drug penetration and increasing the efficacy of topical therapies [16]. Cyclomethicone, an oily ingredient, was commonly included in all L-HSs to ameliorate wetting and spreading of each formula onto the toenail. This oily ingredient has proven to effectively improve the spreading efficiency of topical preparations, primarily by lowering the surface tension between the nail plate and HS [12,13]. To prevent color changes in EFN-loaded topical preparations by oxidation reaction [13], antioxidants, including BHT and EDTA, were included in all preparations.

All L-HS preparations were transparent or pale yellow, with no drug precipitation. The drug content in all L-HS preparations was determined to be between 98% and 105%, with no degradation during the preparation procedure (data not shown). After topical application of a drop of HS base onto the hoof slice, it was rapidly evaporated and/or absorbed into the nail within 14 min (Table 1). On the other hand, the incorporation of oily absorption enhancers markedly lengthened the complete absorption/evaporation time of L-HSs, displaying an absorption/evaporation time ranging from 45.0 to 118.3 min. Despite the rapid vanishing of the organic solvent, non-volatile ingredients, including oily absorption enhancers, remained onto the hoof membrane and were slowly absorbed.

The in vitro permeation profile of EFN through bovine hoof slices after topical application of L-HSs is presented in Figure 2. To guarantee sink conditions during the in vitro permeation study, minimal amount of SLS, an anionic surfactant, was added to the receptor medium (0.2% *w*/*v*)**.** Although sink condition may not occur in vivo, due to the low interstitial fluid volume and to its low renovation rate of nail plate, it was required to compare permeation-enhancing capacity of each formulation more distinctly. In our preliminary test, SLS was effective in increasing the solubility of the sparingly soluble antifungal agent by forming micellar structures in aqueous media above the critical micellar concentration. The solubility of EFN in 0.2% SLS-containing PBS solution was 2.1 mg/mL, whereas the drug solubility in PBS was approximately 0.02 mg/mL (data not shown). After topical application of HS base or L-HSs, drug molecules penetrated the hoof slice, with no marked lag time. Subsequently, the extent of EFN permeated steadily increased for 20 h in all HSs under sink conditions.

The overall amount of EFN permeated was higher in L-HS formulations, including L-HS3 (Labrafac PG, flux value of 138.1 μg∙cm^−2^·h^−1^), L-HS2 (Lauroglycol 90, 129.9 μg∙cm^−2^·h^−1^), L-HS4 (medium-chain triglyceride, 120.8 μg∙cm^−2^·h^−1^), and L-HS1 (marketed formula, 112.0 μg∙cm^−2^·h^−1^), compared to HS base (97.6 μg∙cm^−2^·h^−1^); the amount of EFN permeation after topical application of L-HSs was over 41% (L-HS3), 33% (L-HS2), 24% (L-HS4), and 13% (L-HS5) higher than that of the HS base after 20 h, respectively (Table 4). In particular, L-HS3 containing Labrafac PG as a lipophilic enhancer exhibited the highest permeability profile over the experimental period, showing 36% higher permeation than the commercialized composition (L-HS1) after 20 h. Labrafac PG is an oily ingredient with hydrophilic-lipophilic balance (HLB) value of 2.0, which has been employed as an oil vehicle in topical formulations as an effective solubilizer. In fact, the oily ingredient showed high solubility for EFN (over 200 mg/mL, data not shown). This high solubilizing capacity enabled EFN to maintain the dissolved state even after ethanol evaporation, providing an opportunity for continuous drug penetration. Haque et al. (2017) reported that considerable quantities of Labrafac PG were upheld in the skin tissue after topical application, which was helpful in maintaining water-insoluble drug in dissolved state in the relevant layer [21]. Salimi et al. (2016) revealed that Labrafac PG drastically diminished the skin resistance to drug penetration by obstructing lipid packing in the layer [22]. Taken together, Labrafac PG was included in HS formula as lipophilic enhancer.

### 2.2. Combination of Hydrophilic Enhancer with Lipophilic Enhancer for Transungual EFN Delivery

LH-HS formulas containing hydrophilic enhancers (SLS, DEPC, and HP-β-CD), along with Labrafac PG as a lipophilic enhancer, were further established, and their physicochemical properties and transungual permeation profiles were evaluated (Table 4). As hydrophilic absorption enhancers, several organic and/or inorganic substances have been attempted, such as SLS, HP-β-CD, phosphatidyl choline, urea, pantothenic acid, ascorbic acid, polyethylene glycol 300, potassium phosphate, and N-acetylcysteine [20,23,24]. Among them, pantothenic acid and ascorbic acid were excluded, as they were insoluble or rapidly degraded in HS preparations, respectively. Thiols, including N-acetylcysteine, were not employed, as they might cause irreversible damage to nail plates. In designing LH-HSs, the proportion of distilled water in topical solutions was increased to 5% (*w*/*v*) to solubilize water-soluble enhancers. The increased water content in HS might be beneficial for ungual delivery, as the content of water in the nail plate is 10–30% and as the supplement of aqueous vehicle is important for swelling, elasticity, and flexibility of the nail structure [25]. In contrast, cyclomethicone, the wetting agent, was excluded from the LH-HSs, as this oily ingredient was phase-separated, as the content of hydrophilic substances was increased in preparations. All LH-HS formulations listed in Table 2 (LH-HS1∼LH-HS4) were transparent or pale yellow, with no drug precipitation. The drug content in each formulation was set to 10%, respectively. The time required for absorption and/or evaporation after topical application of LH-HSs (LH-HS1-LH-HS4) was determined to be between 46 and 75 min (Table 3), which is comparable to that of L-HS containing only Labrafac PG as a lipophilic enhancer (L-HS3).

The in vitro permeation profile of the antifungal agent through the hoof slice after topical application of LH-HSs is depicted in Figure 3. The permeation pattern of LH-HSs was analogous with that of L-HSs, exhibiting prompt penetration upon topical application, followed by a gradual increase for 20 h. Among LH-HSs with different absorption enhancers, LH-HS3 containing HP-β-CD (3% *w*/*v*) in combination with Labrafac PG showed higher flux values than LH-HS1 (SLS, 3% *w*/*v*) or LH-HS2 (DEPC, 3% *w*/*v*); the flux values of EFN penetrated for 20 h after the application of LH-HS1, LH-HS2, and LH-HS3 were estimated to be 115.0, 81.8, and 161.0 μg∙cm^−^^2^·h^−^^1^, respectively (Table 4). The permeation of EFN through the hoof slice was further increased as the proportion of HP-β-CD increased (6% *w*/*v*, LH-HS4), providing a flux value of 200.2 μg∙cm^−^^2^·h^−^^1^. The extent of EFN permeation after treatment with LH-HS4 (HP-β-CD, 3% *w*/*v*) was even higher than that with L-HS1 (patented formula) or L-HS3 (Labrafac PG), exhibiting over 78% or 45% higher drug permeation across the hoof membrane, respectively (Table 4). The absorption improvement by HP-β-CD was consistent with the previous report where cyclodextrin derivatives markedly improved the hydration of nail plates and increased the solubility of terbinafine, facilitating drug permeation through the water-filled porous structure of hydrated nail plates [20]. From these findings, we concluded that the combined use of Labrafac PG and HP-β-CD is advantageous in improving the absorption of EFN, with different permeation-facilitating mechanisms.

### 2.3. Effect of Oil Solution for Transungual EFN Delivery

Different EFN-loaded OSs based on Labrafac PG (OS1), peppermint oil (OS2), and almond oil (OS3) were constructed and their physicochemical properties and in vitro permeation profile were further assessed (Table 4). Labrafac PG was confirmed to improve transungual delivery of EFN when included in ethanol-based L-HS2 formula. Essential oils, including peppermint oil or almond oil, are commonly employed in topical preparations for curing toenail fungus, as tocopherol included in essential oils possess antioxidant, anti-inflammatory, and immune-boosting activities [26,27,28]. Moreover, oily vehicles consisting of benzyl alcohol, peppermint oil, turpentine, and mineral oil, effectively promoted the transungual delivery of ciclopirox [29]. Along with oily vehicles, ethanol and distilled water were included to improve the spreadability of the oily solution onto the nail plate. All OS preparations (OL1–OL3) were transparent or pale yellow, with a drug concentration of 10% *w*/*v*. After topical application of OS2, volatile peppermint oil-based solution was completely absorbed and/or evaporated within 30 min. On the other hand, it took about 70 or 176 min after the application of non-volatile oil-based OS1 or OS2, respectively (Table 4).

The in vitro permeation profile of EFN after topical application of OS formulations (OS1–OS3) is shown in Figure 4. OS formulations showed an overall lower permeation profile over the experimental period compared with L-HSs or LH-HSs, with no marked difference depending on the oily vehicle. The flux values of OS1, OS2, and OS3 were 93.0, 79.5, and 71.0 μg∙cm^−^^2^·h^−^^1^, respectively (Table 4). The relatively poor wettability and/or spreadability of OSs might hinder the penetration of EFN-loaded oily vehicles into the hoof slice, showing a lower flux value than HSs, including LH-HS4 or even HS base. LH-HS4 containing both Labrafac PG and HP-β-CD as lipophilic and hydrophilic permeation enhancers, respectively, was chosen as the optimal preparation for the ungual delivery of EFN.

### 2.4. Physicochemical Stability of Optimized EFN-Loaded LH-HS Preparation

The physicochemical stability of the HL-HS4 formula was further evaluated under stress (60 °C for six days) or ambient (25 °C for 10 months) storage conditions (Table 5). The hydroalcoholic preparation of the antifungal agent has been reported to change from transparent to yellowish by oxidation reaction [12]. However, under both stress and ambient conditions, no changes in color and/or drug precipitation were observed in the HL-HS4 formulation. After 10 months storage at 25 °C, drug content in HL-HS4 was determined to 94.8%, with no pH change in liquid preparation. Drug content in HL-HS4 has decreased by 2.5% for 10 months, compared to the initial drug content. Thus, the HL-HS4 formulation containing Labrafac PG and HP-β-CD was assumed to be physicochemically stable, providing over 90% drug content over 24 months under ambient condition.

## 3. Materials and Methods

### 3.1. Materials

EFN powder (purity > 99.9%) was obtained from Cipla Limited (Mumbai, India). Myristyl lactate, diisopropyl adipate, HP-β-CD, SLS, peppermint oil, almond oil, BHT, ethanol, EDTA, sodium azide, phosphate-buffered saline (PBS) tablets, and citric acid were purchased from Sigma-Aldrich Chemical Co. (St. Louis, MO, USA). Labrafac PG, Lauroglycol 90, medium-chain triglyceride (MCT, Labrafac Lipophile WL1349), isopropyl myristate, and isopropyl palmitate were obtained from Gattefosse (Saint-Priest, France). Cyclomethicone and DEPC were acquired from Dow silicone Co. (Midland, MI, USA) and Lipoid (Newark, NJ, USA), respectively.

### 3.2. Preparation of EFN-Loaded Topical Solutions

Different HS preparations containing EFN (10% *w*/*w*) were prepared using a conventional procedure as described previously [23]. The compositions of the HS base and L-HS, and LH-HS prepared in our study are listed in Table 1 and Table 2, respectively. First, EFN (1 g) was added to ethanol (5.3–6.8 g) and stirred for 30 min to obtain a transparent solution. Then, lipophilic permeation enhancers (myristyl lactate, diisopropyl adipate, Labrafac PG, MCT, Lauroglycol 90, isopropyl myristate, and isopropyl palmitate, 1.5–2.2 g), cyclomethicone (0–1.3 g), and BHT (0.01 g) were added to the ethanol solution. Then, aqueous solution (0.1–1.2 g) containing hydrophilic enhancers (HP-β-CD, DEPC, and SLS), EDTA, citric acid, and BHT were added to the drug solution. The mixture was stirred for 10 min using a stirrer. The final weight of the topical solution was adjusted to 10 g by dropwise addition of ethanol.

EFN-loaded OSs with different oily vehicles were further prepared using a conventional procedure, as described previously (Table 3). EFN (1 g) was dissolved in 8 g of oily vehicle (Labrafac PG, almond oil, and peppermint oil) by vigorous vortexing for 30 min. To reduce the viscosity of the drug-loaded oil solution, 0.89 g of ethanol and 0.01 g of distilled water were added to the solution and stirred for 30 min. The prepared formulations were kept in glass vials at room temperature.

### 3.3. Physicochemical Characteristics of EFN-Loaded Topical Preparations

The physicochemical properties of EFN-loaded topical solutions were characterized by appearance, drug content, pH, and absorption/evaporation time into the bovine hoof membrane. After placing the sample in a glass vial, the transparency and the presence of drug precipitates and/or aggregates were visually observed. The acidity of each sample was determined using a pH meter (Mettler Toledo, Greifensee, Switzerland).

The drug content in each formula was determined using HPLC analysis previously reported, with slight modifications [30]. Each sample (100 µL) was 1,000-fold diluted with acetonitrile (ACN) and injected into a Shimadzu HPLC system (Shimadzu, Kyoto, Japan) equipped with a pump (LC-20AD XR), autosampler (SIL-20AC XR), UV–VIS detector SPD-M20A), and system controller (CBM-20A). Chromatographic separation of EFN was achieved isocratically using a reverse-phase C18 column (4.6 mm I.D. × 250 mm, 5 µm, Shiseido, Japan) with a mixture of ACN and distilled water (8:2 v/v) as the mobile phase. The flow rate of the mobile phase was set to 1.0 mL/min. The temperature of the column oven (CTO-20A) was set to 30 °C. The eluent was monitored at 210 nm with a retention time of about 5 min. The calibration curve for EFN was linear in the concentration range of 1–100 μg/mL (y = 36319x + 21103, *r*^2^ = 1.0, where x and y are the drug concentration (μg/mL) and peak area, respectively). The limit of detection (LOD) and the limit of quantitation (LOQ) values were calculated to 0.1 and 0.5 μg/mL, respectively.

### 3.4. Absorption/Evaporation Time of EFN-Loaded Topical Preparations

To compare the absorption and/or evaporation rate of each formula, the absorption/evaporation time of each sample into bovine hoof slices was further evaluated. Before the experiment, bovine hoof tissues were carefully washed and hydrated in distilled water for 24 h. The softened tissues were cut into thin slices using a microtome (RM2245 Semi-Automated Rotary Microtome, Leica, Germany). The thickness of the slices was measured using a micrometer (Mitutuyo, Guipúzcoa, Spain), and slices with similar thickness and no visible faults or fissures were chosen for the study. The thickness of the hoof slices was ranged from 0.20 to 0.25 mm. A drop of sample (approximately 30 μL) was loaded onto a bovine hoof slice using a pipette, and the time required for complete absorption was determined by dry-to-touch in a shaking incubator maintained at 32 °C.

### 3.5. In Vitro Permeation Study of EFN-Loaded Topical Preparations

The in vitro drug permeation profile through bovine hoof slices after topical application of EFN-loaded preparations was evaluated in the vertical Franz diffusion cell model [23,31,32]. On the day of the permeation experiment, the hoof slices immersed in PBS for 1 h were clamped between two Teflon cylindrical adapters (silicone, Vollmond, Seoul, Korea), providing an effective diffusional area of 0.28 cm^2^. Adapters with samples were clamped between the donor and receptor chambers of vertical Franz diffusion cells. The receptor compartment was filled with 12 mL of 10 mM PBS (pH 7.4) containing 0.01% (*w*/*v*) sodium azide as a microbial growth inhibitor and 0.2% (*w*/*v*) SLS as solubility enhancer. Afterward, each formulation containing an equivalent amount of EFN (3 mg) was applied on the donor compartment. During the experiment, the receptor medium was stirred at 450 rpm using a magnetic stirrer, and the temperature was kept at 32 °C. At a predetermined time, 0.2 mL of receptor media was withdrawn using a syringe, and prewarmed fresh media were refurnished in the receptor compartment. The withdrawn samples were diluted with methanol and analyzed using HPLC. Flux value of each formulation was calculated by dividing the accumulative amount of EFN permeated at 20 h (mg/cm^2^) by the time (20 h). The permeation data were statistically analyzed by Student’s *t*-test using SPSS software 17.0.

### 3.6. Physicochemical Stability of EFN-Loaded Topical Preparation

For selected topical preparation, the physicochemical stability was evaluated under stress (60 °C) and ambient conditions (25 °C). LH-HS4 kept in glass vials was stored at 60 °C for 6 days or at 25 °C for 10 months. The withdrawn sample was evaluated by aspect of appearance (phase separation or drug precipitation), acidity, and drug content.

## 4. Conclusions

In this study, EFN-loaded HS preparations containing different lipophilic and/or hydrophilic absorption enhancers were formulated, and their permeation profiles were successfully evaluated using bovine hoof slices. The optimized HS formula combining lipophilic (Labrafac PG) with hydrophilic penetration enhancer (HP-β-CD) offered favorable permeation profile, providing over two-fold increased flux value compared with the HS base with no permeation enhancer. Moreover, the novel formula was physicochemically stable, with no changes in appearance, acidity, and drug content under stress or ambient conditions. Therefore, this novel HS can be a promising tool for the delivery of EFN in the treatment of toenail onychomycosis.

## Figures and Tables

**Figure 1 molecules-26-01650-f001:**
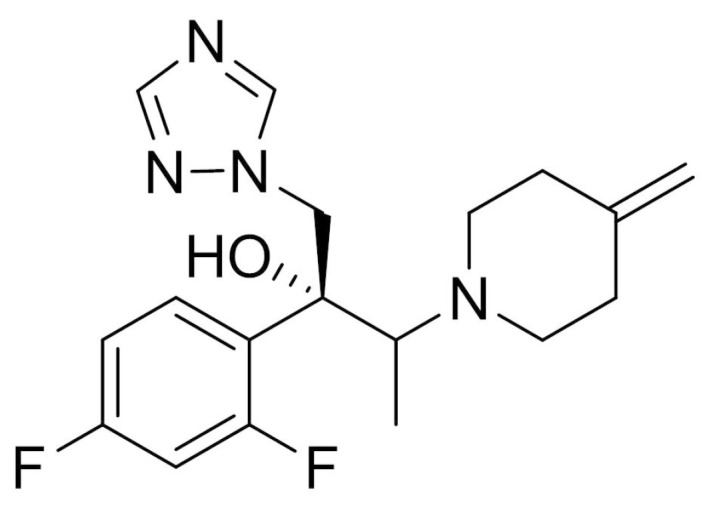
Chemical structure of efinaconazole.

**Figure 2 molecules-26-01650-f002:**
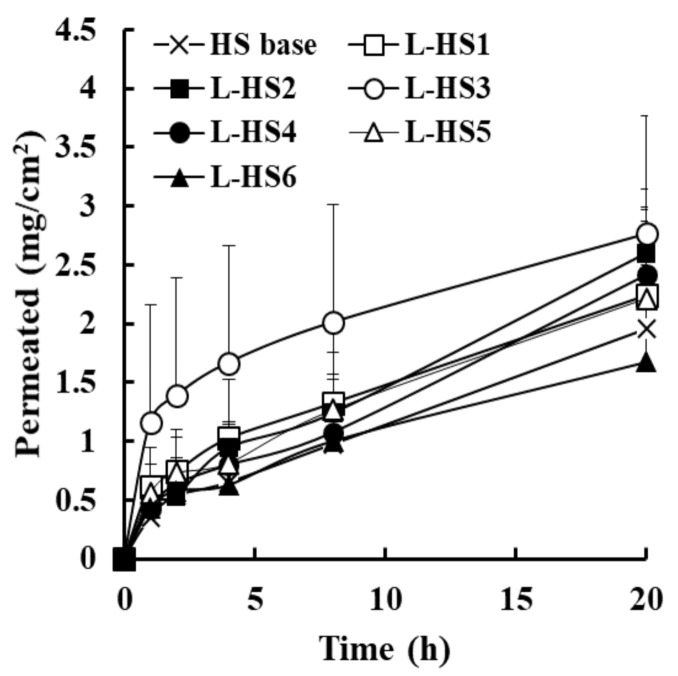
In vitro permeation profile of efinaconazole through the bovine hoof from L-hydroalcoholic solutions (HSs) with different lipophilic enhancers in the Franz diffusion cell model. Data represent mean ± standard deviation (*n* = 4).

**Figure 3 molecules-26-01650-f003:**
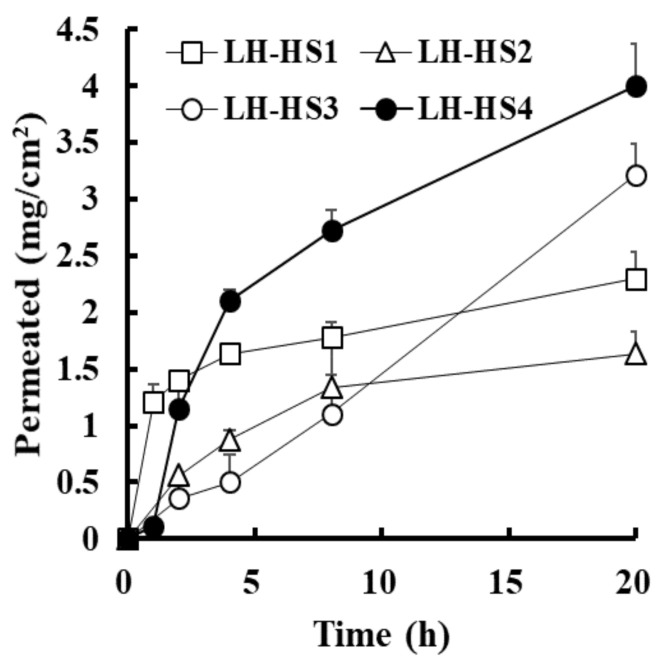
In vitro permeation profile of efinaconazole through the bovine hoof from LH-HS preparations prepared with different hydrophilic enhancers, along with Labrafac PG as a lipophilic enhancer in the Franz diffusion cell model. Data represent mean ± standard deviation (*n* = 4).

**Figure 4 molecules-26-01650-f004:**
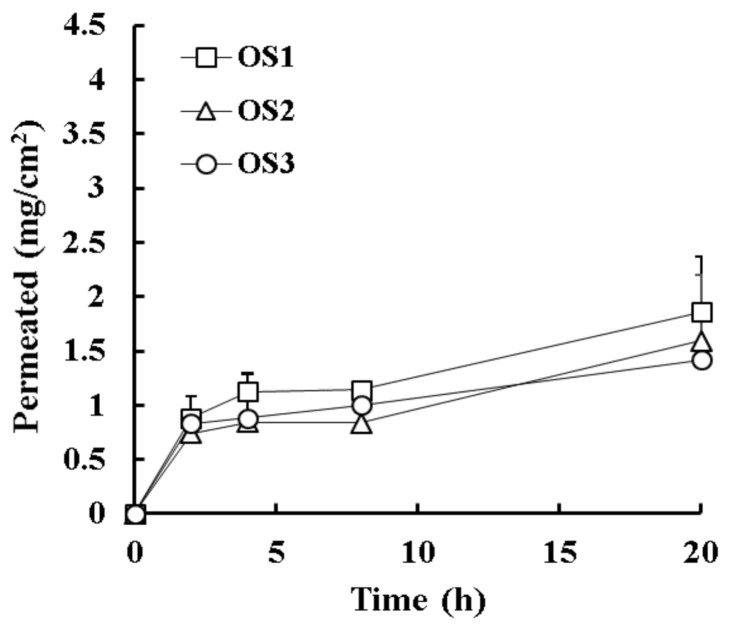
In vitro permeation profile of efinaconazole through the bovine hoof from oil solutions (OSs) prepared with different oily vehicles in the Franz diffusion cell model. Data represent mean ± standard deviation (*n* = 3).

**Table 1 molecules-26-01650-t001:** Compositions and physicochemical characteristics of efinaconazole (EFN)-loaded hydroalcoholic solutions (HSs) with different lipophilic permeation enhancers.

	HS base	L-HS1	L-HS2	L-HS3	L-HS4	L-HS5	L-HS6
*Compositions*							
EFN (g)	1.0	1.0	1.0	1.0	1.0	1.0	1.0
Myristyl lactate (g)	-	1.0	-	-	-	-	-
Diisopropyl adipate (g)	-	1.2	-	-	-	-	-
Lauroglycol 90 (g) ^(1)^	-	-	2.2	-	-	-	-
Labrafac PG (g) ^(2)^	-	-	-	2.2	-	-	-
Medium-chain triglyceride (g)	-	-	-	-	2.2	-	-
Isopropyl myristate (g)	-	-	-	-	-	2.2	-
Isopropyl palmitate (g)	-	-	-	-	-	-	2.2
Cyclomethicone (g)	1.3	1.3	1.3	1.3	1.3	1.3	1.3
EDTA (g) ^(3)^	0.000025	0.000025	0.000025	0.000025	0.000025	0.000025	0.000025
Distilled water (g)	0.1	0.1	0.1	0.1	0.1	0.1	0.1
Citric acid (g)	0.01	0.01	0.01	0.01	0.01	0.01	0.01
BHT (g) ^(4)^	0.01	0.01	0.01	0.01	0.01	0.01	0.01
Ethanol (g)	6.82	5.38	5.38	5.38	5.38	5.38	5.38
Total (g)	10.0	10.0	10.0	10.0	10.0	10.0	10.0
*Physicochemical characteristics*							
Appearance	Transparent	Transparent	Transparent	Transparent	Transparent	Transparent	Transparent
pH ^(5)^	5.1 ± 0.0	4.8 ± 0.1	5.5 ± 0.1	5.2 ± 0.2	5.1 ± 0.00	4.9 ± 0.0	5.2 ± 0.0
Absorption/evaporation time ^(5)^	13.7 ± 0.5	60.0 ± 4.1	75.0 ± 12.2	70.0 ± 4.1	118.3 ± 8.5	93.3 ± 2.4	45.0 ± 2.4

^(^^1)^ Propylene glycol monolaurate; ^(^^2)^ Propylene glycol dicaprylocaprate; ^(^^3)^ Ethylenediaminetetraacetic acid; ^(^^4)^ Butylated hydroxytoluene; ^(^^5)^ Data represent mean ± standard deviation (*n* = 3).

**Table 2 molecules-26-01650-t002:** Compositions and physicochemical characteristics of efinaconazole (EFN)-loaded hydroalcoholic solutions (HSs) with different hydrophilic permeation enhancers, along with Labrafac PG ^(1)^ as a lipophilic enhancer.

	LH-HS1	LH-HS2	LH-HS3	LH-HS4
*Compositions*				
EFN (g)	1.0	1.0	1.0	1.0
Labrafac PG (g)	1.5	1.5	1.5	1.5
SLS (g)	0.3	-	-	-
DEPC (g) ^(^^2)^	-	0.3	-	-
HP-β-CD (g)	-	-	0.3	0.6
EDTA (g)	0.000025	0.000025	0.000025	0.000025
Distilled water (g)	1.2	1.2	1.2	1.2
Citric acid (g)	0.01	0.01	0.01	0.01
BHT (g)	0.01	0.01	0.01	0.01
Ethanol (g)	5.78	5.78	5.78	5.48
Total (g)	10.0	10.0	10.0	10.0
*Physicochemical characteristics*				
Appearance	Transparent	Transparent	Transparent	Transparent
pH ^(^^3)^	6.7 ± 0.0	5.2 ± 0.0	5.2 ± 0.1	5.2 ± 0.1
Absorption/evaporation time ^(^^3)^	46.0 ± 2.9	54.0 ± 2.9	75.0 ± 4.1	70.0 ± 4.1

^(^^1)^ Propylene glycol monolaurate; ^(^^2)^ 1,2-dielaidoyl-sn-glycero-3-phosphocholine; ^(^^3)^ Data represent mean ± standard deviation (*n* = 3).

**Table 3 molecules-26-01650-t003:** Compositions and physicochemical characteristics of efinaconazole (EFN)-loaded oil solutions (OSs) prepared with different oily vehicles.

	OS1	OS2	OS3
*Compositions*			
EFN (g)	1.0	1.0	1.0
Labrafac PG (g)	8.0	-	-
Peppermint oil (g)	-	8.0	-
Almond oil (g)	-	-	8.0
Distilled water (g)	0.01	0.01	0.01
Ethanol (g)	0.89	0.89	0.89
Total (g)	10.0	10.0	10.0
*Physicochemical* *characteristics*			
Appearance	Transparent	Transparent	Transparent
pH ^(1^^)^	9.0 ± 0.0	6.0 ± 0.0	6.2 ± 0.0
Absorption/evaporation time ^(^^1)^	71.0 ± 2.9	27.3 ± 2.1	176.7 ± 4.7

^(1)^ Data represent mean ± standard deviation (*n* = 3).

**Table 4 molecules-26-01650-t004:** Flux and accumulated amount of efinaconazole (EFN) permeated through the bovine hoof for 20 h after the topical application of hydroalcoholic solutions (HSs) or oil solutions (OSs) in Franz diffusion cell model.

Formulas	Flux (μg∙cm^−2^·h^−1^)	Permeated (%)
HL base	97.6 ± 27.5	18.2 ± 5.1
L-HS1	112.0 ± 45.0	20.9 ± 8.4
L-HS2	129.9 ± 18.6	24.3 ± 3.5
L-HS3	138.1 ± 35.0	25.8 ± 6.5
L-HS4	120.8 ± 22.8	22.6± 4.2
L-HS5	110.5 ± 38.9	20.6 ± 7.3
L-HS6	83.9 ± 47.6	15.7± 8.9
LH-HS1	115.0 ± 5.8	21.4 ± 2.2
LH-HS2	81.8 ± 6.67	15.3 ± 1.8
LH-HS3	161.0 ± 8.6 *^,^ **	30.3 ± 2.6 *^,^ **
LH-HS4	200.2 ± 8.9 *^,^ **	37.4 ± 3.5 *^,^ **
OS1	93.0 ± 1.5	17.4 ± 0.3
OS2	79.5 ± 38.5	14.8 ± 7.2
OS3	71.0 ± 39.0	13.3± 7.3

Data represent mean ± standard deviation (*n* = 3–4). * *p* < 0.05 versus HS-base; ** *p* < 0.05 versus marketed composition (L-HS1).

**Table 5 molecules-26-01650-t005:** Physicochemical stability of the optimized topical solution (LH-HS4) under stress (60 °C for six days) and ambient condition (25 °C for 10 months).

	After Preparation	Under Stress Conditions	Under Ambient Conditions
Appearance	Transparent	Transparent	Transparent
pH ^(1)^	5.2 ± 0.1	5.2 ± 0.0	5.3 ± 0.01
Drug content (%) ^(1)^	97.3 ± 1.8	94.0 ± 1.1	94.8 ± 0.0

^(1)^ Data represent mean ± standard deviation (*n* = 3).

## Data Availability

The data that support the findings of this study are available from the corresponding author upon reasonable request.

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
