# Peer review of "Effect of Penetration Enhancers on Toenail Delivery of Efinaconazole from Hydroalcoholic Preparations"

_molecules, 2021, doi:10.3390/molecules26061650_

Round 1
Reviewer 1 Report
The manuscript covers an interesting topic. It is well organized, tables and diagrams give an appropriate overview of the findings. There are just some minor points that may be improved.
General: please specify in the introduction what is the therapeutic gap that shall e closed by the new formulation. Or what´s wrong with the marketed formulation? Why does it need to be improved?
As I understand, the authors did also compare their formulations with the marketed one. It would be good to point this out more specifically.
line 135 and 138: 41%, 33% or 36% - which was the highest enhancement and which formulation does this correspond to? Please make this clear in the text.
line 139: sentence starts with "Labrafac.." but it is not clear which of the formulations this refers to (yes, composition is given in table 1 but it would be easier to follow the text if a hint was given in the text and one would not have to look it up in the table)
stability: 5% degradation in 10 months would make a shelf life of 10 months, which is to short for a makreted formulation. The authors should add a comment on this in the text.
HPLC method: LOD & LOQ should be added
thickness of hoof slices wa 0.4-0.8 mm: this is a difference of 100%. What is the impact on the permeation results? How was this accounted for?
SLS in receptor medium: SLS is known to enhance the permation of many drugs though skin. What is the situation in nail and what effect does the use of SLS in the receptor medium hve on the permeation of EFN?
Author Response
[For Comments by Reviewer 1]
Thanks for reviewing the manuscript in detail: Ms. Ref. No.: Molecules. 1060158. The reviewer's comments were carefully studied and reflected in the revised manuscript. We tried our best to follow your precious commentary. Please find the separate sheets attached which addresses, point-by-point, the issues raised by the reviewers.
- Please specify in the introduction what is the therapeutic gap that shall enclosed by the new formulation. Or what´s wrong with the marketed formulation? Why does it need to be improved?
à Thanks for your invaluable comment. In accordance your comment, the need to formulate new topical preparation with facilitated permeation profile is further described in the manuscript as follows (See page 2, line 51): “In designing topical preparations of antifungal agents, especially for toenail treatment, endowing satisfactory penetration into relevant layer, including the nail plate, is one of the pivotal factors influencing therapeutic efficacy [7,8,9].
Also see page 2, line 63, “Actually, the increased topical delivery of EFN markedly accelerated the structural recovery of keratin layer in a guinea pig onychomycosis model [13].”
[13] Lee, B.C.; Pangeni, R.; Na, J.; Koo, K.T.; Park. J.W. Preparation and in vivo evaluation of a highly skin- and nail-permeable efinaconazole topical formulation for enhanced treatment of onychomycosis. Drug Deliv. 2019, 26, 1167–1177
- line 135 and 138: 41%, 33% or 36% - which was the highest enhancement and which formulation does this correspond to? Please make this clear in the text.
à Thanks for your invaluable comment. In accordance your comment, formulation code was further described in the sentence to express the data more clearly (please see page 6, line 158): “the amount of EFN permeation after topical application of L-HSs was over 41% (L-HS3), 33% (L-HS2), 24% (L-HS4), and 13% (L-HS5) higher than that of the HS base after 20 h, respectively (Table 4).”
- line 139: sentence starts with "Labrafac.." but it is not clear which of the formulations this refers to (yes, composition is given in table 1 but it would be easier to follow the text if a hint was given in the text and one would not have to look it up in the table)
à Thanks for your invaluable comment and sorry for our insufficient description. The sentence “Labrafac PG is an oily ingredient with hydrophilic-lipophilic balance (HLB) value of 2.0, which has been employed as an oil vehicle in topical formulations as an effective solubilizer” was not a description on our specific formula. Thus, formulation code (L-HS3) was not marked in this sentence. Alternatively, we described that Labrafac PG was contained in L-HS3 in the previous sentence as follows (please see page 6, line 160): “In particular, L-HS3 containing Labrafac PG as a lipophilic enhancer exhibited the highest permeability profile over the experimental period, showing 36% higher permeation than the commercialized composition (L-HS1) after 20 h.”
- stability: 5% degradation in 10 months would make a shelf life of 10 months, which is to short for a makreted formulation. The authors should add a comment on this in the text.
à Thanks for your invaluable comment. In accordance to your comment, further discussion on drug stability was described in the manuscript as follows (see page 8, line 265): “After 10 months storage at 25 °C, drug content in HL-HS4 was determined to 94.8%, with no pH change in liquid preparation. Drug content in HL-HS4 has decreased by 2.5% for 10 months, compared to the initial drug content. Thus, the HL-HS4 formulation containing Labrafac PG and HP-β-CD was assumed to be physicochemically stable, providing over 90% drug content over 24 months under ambient condition.”
- HPLC method: LOD & LOQ should be added.
à Thanks for your invaluable comment. In accordance to your comment, the information about LOD and LOQ of HPLC assay was described in the manuscript as follows (See page 9, line 318): “The calibration curve for EFN was linear in the concentration range of 1–100 μg/mL (y = 36319 x + 21103, r2 = 1.0, where x and y are the drug concentration (μg/mL) and peak area, respectively). The limit of detection (LOD) and the limit of quantitation (LOQ) values were calculated to 0.1 and 0.5 μg/mL, respectively.”
- Thickness of hoof slices wa 0.4-0.8 mm: this is a difference of 100%. What is the impact on the permeation results? How was this accounted for?
à Thanks for your critical comment and very sorry for our typo. The thickness of hoof slices employed in our study was 0.20-0.25 mm, with minimal difference. The typo was corrected in the manuscript as follows (see page 10, line 329): “The thickness of the hoof slices was ranged from 0.20 mm to 0.25 mm.”
- SLS in receptor medium: SLS is known to enhance the permation of many drugs though skin. What is the situation in nail and what effect does the use of SLS in the receptor medium have on the permeation of EFN?
à Thanks for your invaluable comment. As you mentioned, SLS is known to enhance the permeability of therapeutic agents into skin or nail plate. Nevertheless, in our study, minimal amount of SLS (0.2 w/v%) was included to receptor medium to dissolve the drug permeated. According to the previous report (Cutrín-Gómez et al., 2018), SLS significantly improved nail porosity and permeability of ciclopirox olamine into nail plate at high concentration (5.0% w/v), whereas at low concentration (0.5% w/v), it did not markedly affect on nail porosity and the drug permeability. Thus, we assumed that 0.2% w/v of SLS employed in receptor media might have minimal effect on permeation profile of EFN across the nail plate. Thanks again for your comment.
Cutrín-Gómez, E.; Anguiano-Igea, S.; Delgado-Charro, M.B.; Gómez-Amoza, J.L.; Otero-Espinar, F.J. Effect of penetration enhancers on drug nail permeability from cyclodextrin/poloxamer-soluble polypseudorotaxane-based nail lacquers. Pharmaceutics. 2018, 10, 273.
Reviewer 2 Report
NOTE: This reviewer requires a clear identification of the parts of the manuscript altered by the authors as a result of the comments below.
This is a well organized and well-written manuscript, concerning a study aiming to enhance the penetration of EFN into toenails, for the treatment of onychomycosis. It does so employing lipophilic and hydrophilic penetration enhancers, studying the penetration of EFN in a bovine hoof model.
Major issues:
- This study employs a drug used in onychomycosis treatment. As such, it would be important to have some brief considerations about this condition, in particular, its epidemiology and etiology.
- An animal model of the human toenail - the bovine hoof - is employed. Despite being a common toenail model, considerations about the differences between this model and the human toenail should be present in the manuscript since, in particular, the animal hoof has an higher hydration capacity, is less crosslinked and more permeable than the human toenail, which has implications in the extrapolation to the toenail of the results obtained with this model.
- In the permeation study, a model based on the Franz diffusion cell was employed. The acceptor medium selected as a surrogate of the toenail extracellular fluid was PBS with SLS and sodium azide, at pH 7.4 and at 32 ºC. The choice of this pH value should be briefly justified, since the pH of the human nail surface has been reported to be around 5 (surface) and around 4 (interior). Additionally, the temperature selected as the typical temperature of the toenail should be supported by a reference.
- The authors employed sink conditions in the permeation study. Although it is customary to do so, sink conditions may not occur in vivo, due to the low interstitial fluid volume and to its low renovation rate, as well as to the poor solubility of the drug. The authors should briefly mention this aspect of their model.
- The type of assay employed by the authors to determine what they call the “absorption times” of the formulations (in the tables) and “absorption and/or evaporation” times (in the text), does not allow to separate absorption (penetration) from evaporation, as the authors recognize. Distinguishing evaporation from penetration for the different formulations may be easily achieved (for instance, by video recording of the drop profile evolution on the hoof surface and on a surface in which it does not prenetrate). It may aid in the interpretation of the results obtained with the different penetration enhancers, and may allow establishing correlations between formulation composition, permeation rates and penetration and evaporation of the formulations.
- When comparing means, a statistical analysis is required, so that one may know whether the differences between the means have statistical significance. A statistical analysis of the obtained data is missing and must be done.
- As not all the formulations prepared have the same pH value, varying from 4.8 to 9, the authors failed to address the influence of the pH of the formulation on the results obtained, since it will affect both the ionization of EFN and of the nail keratins.
- Table 1: Instead of “absorption time”, it should appear “absorption/evaporation time”, since the authors acknowledge in the text that this assay provides “absorption and/or evaporation“ times.
- On Tables 2 and 3, the authors present the obtained fluxes for the different formulations. However, they have never mentioned how they were calculated and must mention it.
- Although the authors state twice, in the Results section (but not in the Methods section), that the pH of all formulations was adjusted to 5.0, all formulations have different pH values. This aspect needs clarification.
Minor issues:
- Table 1: The authors should mention early in the manuscript that formulation L-HS1 is the commercial formulation, since that is only revealed in the page that comes after Table 1 and it is not mentioned in the Methods section. Otherwise, the reader may wonder why in that formulation two penetration enhancers are employed while all the others have just one.
- Table 2 includes results from formulations LH-HS1 to 4 and OS1 to 3 but, their composition is given long after this table, in Table 3. The composition of these formulations should appear before mentioning Table 2 in the text.
- Table 2 caption: It would be helpful to the reader to mention the animal model and the assay model in this caption, as it was done in the captions of Figures 3 and 4.
- Line 158: The authors use the word “material” to categorize the penetration enhancers mentioned, although only one of them (PEG) may be properly called a material. Another term would be preferable, such as “substance”.
- When abbreviations are employed, it is customary to present the full name when the abbreviation appears for the first time. This does not happen in the case of BHT, EDTA, DEPC, PC, etc. It could also be done in the case of Lauroglycol 90 and Labrafac PG.
- The abbreviations HS and OS appear in full in the Abstract or in tables’ captions only. They must also appear in full in the main text, when used for the first time.
- Table 4: Formulations OL1 to OL3 should be changed to OS1 to OS3.
- Line 1400: The abbreviation HLB was not presented in full.
- In the Abstract, there is a numerical reference (“[1]”) to a publication in the references list. It is not common to have references cited in an Abstract; it should be corrected.
- Line 696: Reference to a publication by Saner et al. is made, employing the author-date style, while reference through numbers in square brackets is used throughout the manuscript. Although the authors may mention the name of the author as they did in this sentence, they must also add the reference number.
- In some places, the authors indicate the number of replicas by “n=4”. They should separate the equal sign from the digits and have the letter “n” in italics (n = 4).
- Page numbers are not correct, since pg. 1 appears three times.
Author Response
Thanks for reviewing the manuscript in detail: Ms. Ref. No.: Molecules. 1060158. The reviewer's comments were carefully studied and reflected in the revised manuscript. We tried our best to follow your precious commentary.
- This study employs a drug used in onychomycosis treatment. As such, it would be important to have some brief considerations about this condition, in particular, its epidemiology and etiology.
à Thanks for your invaluable comment. In accordance to your comment, the epidemiology and etiology of onychomycosis were further described in the manuscript as follows (see page 1, line 31): “Onychomycosis is one of the most chronic and prevalent nail disorders, caused by fungal infection. Epidemiological studies suggest about 19% of the global population is affected by onychomycosis caused by dermatophytes (Trichophyton rubrum and Trichophyton mentagrophytes), non-dermatophyte molds, and yeasts (Candida albicans) [1,2].”
[1]. David de B. Nails and hair. Medicine. 2009, 37, 286-290.
[2]. Sigurgeirsson, B.; Steingrimsson, O. Risk factors associated with onychomycosis. J. Eur. Acad. Dermatol. Venereol. 2004, 18, 48–51.
- An animal model of the human toenail - the bovine hoof - is employed. Despite being a common toenail model, considerations about the differences between this model and the human toenail should be present in the manuscript since, in particular, the animal hoof has an higher hydration capacity, is less crosslinked and more permeable than the human toenail, which has implications in the extrapolation to the toenail of the results obtained with this model.
à Thanks for your invaluable comment. We absolutely agree on your comment that the bovine hoof slices are quite different with human toenail, as the animal hoof has higher hydration capacity, with less crosslinked and more permeable structure. Nevertheless, as the accessibility to human toenail was not easy, the permeation profile was alternatively assessed using hoof slice model, which has established as an alternative. The main purpose of our study was to comparatively evaluate the permeation capacity between permeation enhancers, thus, we assumed that permeation study using hoof slice model might be valid. Nevertheless, we agree that evaluation using human toenail would be better to predict the nail absorption in humans more accurately. Thanks again for the comments.
- In the permeation study, a model based on the Franz diffusion cell was employed. The acceptor medium selected as a surrogate of the toenail extracellular fluid was PBS with SLS and sodium azide, at pH 7.4 and at 32 ºC. The choice of this pH value should be briefly justified, since the pH of the human nail surface has been reported to be around 5 (surface) and around 4 (interior). Additionally, the temperature selected as the typical temperature of the toenail should be supported by a reference.
à Thanks for your invaluable comment. We are sorry that our experimental model based on the Franz diffusion cell was not fully reflect the human nail surface and interior pH circumstance. References on our experimental condition were attached in the manuscript as follows (see page 10, line 335) : “The in vitro drug permeation profile through bovine hoof slices after topical application of EFN-loaded preparations was evaluated in the vertical Franz diffusion cell model [23,32,33].” Thanks again for your comment.
- Cutrín-Gómez, E.; Anguiano-Igea, S.; Delgado-Charro, M.B.; Gómez-Amoza, J.L.; Otero-Espinar, F.J. Effect on nail structure and transungual permeability of the ethanol and poloxamer ratio from cyclodextrin-soluble polypseudorotaxanes based nail lacquer. Pharmaceutics. 2018, 10, 156
- Nogueiras‐Nieto, L.; Gomez‐Amoza, J.L.; Delgado‐Charro, M.B.; Otero‐Espinar, F.J. Hydration and N‐acetyl‐l‐cysteine alter the microstructure of human nail and bovine hoof: implications for drug delivery. J. Control. Release. 2011, 156, 337–344.
- Walter, R.; Edward S.; Murray J. Skin temperature response of normal human subjects to various conditions. Circulation. 1952, 6, 862-867.
- The authors employed sink conditions in the permeation study. Although it is customary to do so, sink conditions may not occur in vivo, due to the low interstitial fluid volume and to its low renovation rate, as well as to the poor solubility of the drug. The authors should briefly mention this aspect of their model.
à Thanks for your invaluable comment. In accordance to your comment, in vivo circumstance in nail plate was further described in the manuscript as follows (See page 5, line 141): “Although sink condition may not occur in vivo, due to the low interstitial fluid volume and to its low renovation rate of nail plate, it was required to compare permeation-enhancing capacity of each formulation more distinctly.”
- The type of assay employed by the authors to determine what they call the “absorption times” of the formulations (in the tables) and “absorption and/or evaporation” times (in the text), does not allow to separate absorption (penetration) from evaporation, as the authors recognize. Distinguishing evaporation from penetration for the different formulations may be easily achieved (for instance, by video recording of the drop profile evolution on the hoof surface and on a surface in which it does not prenetrate). It may aid in the interpretation of the results obtained with the different penetration enhancers, and may allow establishing correlations between formulation composition, permeation rates and penetration and evaporation of the formulations.
à Thanks for your invaluable comment. We absolutely agree your comment that distinguishing evaporation from penetration may helpful to establish correlations between formulation composition, permeation rates and penetration and evaporation of the formulations. However, unfortunately, it is now difficult to conduct additional experiments to distinguish absorption and evaporation, due to corona pandemic and expiration of the project. Alternatively, we could assume that the delayed absorption/evaporation of time of each formula compared to HS free-base might be mainly regarded as the time taken for absorption, as ethanol has high volatility and oily and/or hydrophilic absorption enhancers have poor evaporation property. In the manuscript, the term ‘absorption time’ was corrected to ‘absorption/evaporation time’. Thanks again for your comment.
- When comparing means, a statistical analysis is required, so that one may know whether the differences between the means have statistical significance. A statistical analysis of the obtained data is missing and must be done.
à Thanks for your invaluable comment. In accordance your comment, statistical analysis was carried out on data obtained from permeation study. Analysis method and result were described in the manuscript as follows (see page 10, line 350): “The permeation data were statistically analyzed by Student’s t-test using SPSS software 17.0.”
Also see Tablet 4 (page 6, line 176)
* p < 0.05 versus HS-base
** p < 0.05 versus marketed composition (L-HS1)
- As not all the formulations prepared have the same pH value, varying from 4.8 to 9, the authors failed to address the influence of the pH of the formulation on the results obtained, since it will affect both the ionization of EFN and of the nail keratins.
à Thanks for your invaluable comment. We agreed on your comment that degree of ionization of EFN and nail keratin might be different by pH value of formulations applied. However, although pH of every formulations was not unified, most formulations except for LH-HS1 and OS1 were prepared with pH value between 4.8 and 6.0 Thus, except for HS1 and OS1, the effect of permeation enhancer might be evaluated with no pH effect. Also, pKa value of efinaconazole is known to be about 12.7, so the degree of ionization did not markedly differ between formulations. Thanks again for your comment.
- Table 1: Instead of “absorption time”, it should appear “absorption/evaporation time”, since the authors acknowledge in the text that this assay provides “absorption and/or evaporation“ times.
à Thanks for your critical comment. In accordance to your comment, “absorption time” in the manuscript including Table 1, 2, and 3 was corrected to “absorption and/or evaporation time”.
- On Tables 2 and 3, the authors present the obtained fluxes for the different formulations. However, they have never mentioned how they were calculated and must mention it.
à Thanks for your critical comment. In accordance to your comment, calculation method was further described in the manuscript as follows (See page 10, line 348): “Flux value of each formulation was calculated by dividing the accumulative amount of EFN permeated at 20 h (mg/cm2) by the time (20 h).”
- Although the authors state twice, in the Results section (but not in the Methods section), that the pH of all formulations was adjusted to 5.0, all formulations have different pH values. This aspect needs clarification.
à Thanks for your careful comment. As pH of all formulations was not adjusted to 5, the statement was removed from the manuscript as follows (See page 5, line 128): “The pH of all preparations was adjusted to 5.0 by adding 10 mM of citric acid.
Also see page 7, line 202: “The drug content in each formulation was set to 10%, respectively.”
Minor issues
1) Table 1: The authors should mention early in the manuscript that formulation L-HS1 is the commercial formulation, since that is only revealed in the page that comes after Table 1 and it is not mentioned in the Methods section. Otherwise, the reader may wonder why in that formulation two penetration enhancers are employed while all the others have just one.
à Thanks for your careful review. In accordance to your comment, we described early that formulation L-HS1 is the commercial formulation in the manuscript as follows (See page 2, line 86): “After selecting a lipophilic enhancer, HSs containing both lipophilic and hydrophilic permeation enhancers (LH-HSs, Table 2) were further designed, and their permeation profiles were evaluated with L-HSs including formula similar to the marketed product (L-HS1), or several oil solutions (OSs, Table 3).”
2) Table 2 includes results from formulations LH-HS1 to 4 and OS1 to 3 but, their composition is given long after this table, in Table 3. The composition of these formulations should appear before mentioning Table 2 in the text.
à Thanks for your careful review. In accordance to your comment, Table 2 and 3 containing composition information was moved before Table 4 containing permeation data.
3) Table 2 caption: It would be helpful to the reader to mention the animal model and the assay model in this caption, as it was done in the captions of Figures 3 and 4.
à Thanks for your careful comment. In accordance to your comment, animal model and the assay mode were contained in Table 2 as follows (see page 6, line 176): “Table 4. Flux and accumulated amount of efinaconazole (EFN) permeated through the bovine hoof for 20 h after the topical application of hydroalcoholic solutions (HSs) or oil solutions (OSs) in Franz diffusion cell model.”
4) Line 158: The authors use the word “material” to categorize the penetration enhancers mentioned, although only one of them (PEG) may be properly called a material. Another term would be preferable, such as “substance”
à Thanks for your careful review. In accordance your comment, the term “material” was corrected to “substances” in the manuscript as follows (see page 6, line 188): “As hydrophilic absorption enhancers, several organic and/or inorganic substances have been attempted, such as SLS, HP-β-CD, PC, urea, pantothenic acid, ascorbic acid, polyethylene glycol 300, potassium phosphate, and N-acetylcysteine.”
5) When abbreviations are employed, it is customary to present the full name when the abbreviation appears for the first time. This does not happen in the case of BHT, EDTA, DEPC, PC, etc. It could also be done in the case of Lauroglycol 90 and Labrafac PG.
à Thanks for your careful review. In accordance your comment, the full name of BHT, EDTA, DEPC, PC, Lauroglycol 90 and Labrafac PG, and so on was described in the manuscript for the first time.
6) The abbreviations HS and OS appear in full in the Abstract or in tables’ captions only. They must also appear in full in the main text, when used for the first time.
à Thanks for your careful review. In accordance your comment, the full name of HS and OS was described in the Introduction section as follows (see page 2, line 83): “The objective of the present study was to design a topical preparation for EFN with high ungual permeation profile. Different hydroalcoholic solutions (HSs) containing several lipophilic enhancers (L-HSs, Table 1) were formulated, and their physicochemical properties and permeation profile through the bovine hooves were evaluated. After selecting a lipophilic enhancer, HSs containing both lipophilic and hydrophilic permeation enhancers (LH-HSs, Table 2) were further designed, and their permeation profiles were evaluated with L-HSs including formula similar to the marketed product (L-HS1), or several oil solutions (OSs, Table 3).”
7) Table 4: Formulations OL1 to OL3 should be changed to OS1 to OS3.
à Sorry for our typo. In accordance your comment, we changed OL to OS in Table 2.
8) Line 140: The abbreviation HLB was not presented in full.
à Thanks for your careful review. In accordance your comment, the full name of HLB was described in the manuscript as follows (See page 6, line 163): “Labrafac PG is an oily ingredient with hydrophilic-lipophilic balance (HLB) value of 2.0, which has been employed as an oil vehicle in topical formulations as an effective solubilizer.”
9) In the Abstract, there is a numerical reference (“[1]”) to a publication in the references list. It is not common to have references cited in an Abstract; it should be corrected.
à Thanks for your careful review. In accordance your comment, reference in the abstract section was removed.
10) Line 696: Reference to a publication by Saner et al. is made, employing the author-date style, while reference through numbers in square brackets is used throughout the manuscript. Although the authors may mention the name of the author as they did in this sentence, they must also add the reference number.
à Thanks for your careful review and sorry for our mistake. In accordance your comment, reference number was attached (see page 2, line 72): “Saner et al. (2014) reported that topical treatment with lipophilic excipients principally altered the microstructure of the nail plate through physical interaction with the lipid components of the plate, promoting drug penetration [16].”
Also see page 5, line 119 “Saner et al. (2014) reported that these lipophilic ingredients altered the microstructure of the nail plate by interacting with its lipid components, promoting drug penetration and increasing the efficacy of topical therapies [16].”
11) In some places, the authors indicate the number of replicas by “n=4”. They should separate the equal sign from the digits and have the letter “n” in italics (n = 4).
à Thanks for your careful review. In accordance your comment, we separate the equal sign from the digits and have the letter “n” in italics (n = 4).
12) Page numbers are not correct, since pg. 1 appears three times.
à Thanks for your careful comment and sorry for our mistake. Page number error was corrected. Thanks again for your invaluable comments.
Round 2
Reviewer 2 Report
REPLY TO SOME OF THE AUTHOR’S COMMENTS
1. This study employs a drug used in onychomycosis treatment. As such, it would be important to have some brief considerations about this condition, in particular, its epidemiology and etiology.
AUTHOR'S REPLY:
Thanks for your invaluable comment. In accordance to your comment, the epidemiology and etiology of onychomycosis were further described in the manuscript as follows (see page 1, line 31): “Onychomycosis is one of the most chronic and prevalent nail disorders, caused by fungal infection. Epidemiological studies suggest about 19% of the global population is affected by onychomycosis caused by dermatophytes (Trichophyton rubrum and Trichophyton mentagrophytes), non-dermatophyte molds, and yeasts (Candida albicans) [1,2].”[1]. David de B. Nails and hair. Medicine. 2009, 37, 286-290. [2]. Sigurgeirsson, B.; Steingrimsson, O. Risk factors associated with onychomycosis. J. Eur. Acad. Dermatol. Venereol. 2004, 18, 48–51.
REVIEWER COMMENT:
The importance of the work in this manuscript and of onychomycosis would be highlighted if, in addition to the information now added by the authors, some brief information concerning the high prevalence among the elderly, the predisposing factors and the impact in quality of life was also present in the manuscript.
2. An animal model of the human toenail - the bovine hoof - is employed. Despite being a common toenail model, considerations about the differences between this model and the human toenail should be present in the manuscript since, in particular, the animal hoof has an higher hydration capacity, is less crosslinked and more permeable than the human toenail, which has implications in the extrapolation to the toenail of the results obtained with this model.
AUTHOR'S REPLY:
Thanks for your invaluable comment. We absolutely agree on your comment that the bovine hoof slices are quite different with human toenail, as the animal hoof has higher hydration capacity, with less crosslinked and more permeable structure. Nevertheless, as the accessibility to human toenail was not easy, the permeation profile was alternatively assessed using hoof slice model, which has established as an alternative. The main purpose of our study was to comparatively evaluate the permeation capacity between permeation enhancers, thus, we assumed that permeation study using hoof slice model might be valid. Nevertheless, we agree that evaluation using human toenail would be better to predict the nail absorption in humans more accurately. Thanks again for the comments.
REVIEWER COMMENT:
This reviewer’s remark was not about the validity of the animal model selected, since it is fully acceptable. However, when employing an animal model, even with the purpose of comparing different formulations, if the main differences between the model and the human equivalent are not presented, a reader not familiar with this area might be tempted to take the best formulation in the manuscript as the best for the human toenail, when that might not be the case. As such, brief information concerning the main differences between the animal model and the human toenail, that are absent in the manuscript, should be included, perhaps in the final discussion of the results obtained.
3. The type of assay employed by the authors to determine what they call the “absorption times” of the formulations (in the tables) and “absorption and/or evaporation” times (in the text), does not allow to separate absorption (penetration) from evaporation, as the authors recognize. Distinguishing evaporation from penetration for the different formulations may be easily achieved (for instance, by video recording of the drop profile evolution on the hoof surface and on a surface in which it does not prenetrate). It may aid in the interpretation of the results obtained with the different penetration enhancers, and may allow establishing correlations between formulation composition, permeation rates and penetration and evaporation of the formulations.
REPLY:
Thanks for your invaluable comment. We absolutely agree your comment that distinguishing evaporation from penetration may helpful to establish correlations between formulation composition, permeation rates and penetration and evaporation of the formulations. However, unfortunately, it is now difficult to conduct additional experiments to distinguish absorption and evaporation, due to corona pandemic and expiration of the project. Alternatively, we could assume that the delayed absorption/evaporation of time of each formula compared to HS free-base might be mainly regarded as the time taken for absorption, as ethanol has high volatility and oily and/or hydrophilic absorption enhancers have poor evaporation property. In the manuscript, the term ‘absorption time’ was corrected to ‘absorption/evaporation time’. Thanks again for your comment.
COMMENT:
The authors should add those assumptions to the manuscript.
COMMENTS ABOUT THE STATISTICAL ANALYSIS OF THE RESULTS
1. The significance level of the statistical analysis should appear in the Methods section.
2. Assuming that the data follow a Gaussian distribution (in case they don’t, nonparametric tests would have to be employed), when one wants to compare several sample means (the formulations) to a mean taken as a “control” (HS base), a one-way ANOVA must be done, followed by a “post-hoc” multiple comparison test (in this case, Dunnett’s test). The authors have employed multiple Student t tests to compare each mean to the control; this is not correct.
3. The authors have now done a statistical analysis of results of the permeation study but did not do it for the absorption/evaporation (abs/evap) times. In the present form of the manuscript, it may be accepted since, in the discussion of the results, a comparison of the different formulations regarding these abs/evap times is not done, as the authors have just compared HS base with the L-HS formulations and the times are very different. However, they could have compared all the formulations to the HS base formulation, employing a statistical test.
4. Lines 203-206: “The time required for absorption and/or evaporation after topical application of LH-HSs (LH-HS1-LH-HS4) was determined to be between 46 and 75 min (Table 3), which is comparable to that of L-HS containing only Labrafac PG as a lipophilic enhancer (L-HS3).” Here, the authors compare the abs/evap times of the LH-HS formulations. These results appear in Table 2, not in Table 3. Additionally, the authors state that the abs/evap times of the LH-HS formulations (ranging from 46 ± 2.9 to 75 ± 4.1 min) are comparable to that of the L-HS3 (70 ± 4.1; Table 1) but this comparison requires a statistical analysis and the results should appear in the manuscript (employing asterisks).
5. Lines 210 – 220: “Among LH-HSs with different absorption enhancers, LH-HS3 containing HP-β-CD (3% w/v) in combination with Labrafac PG showed higher flux values than LH-HS1 (SLS, 3% w/v) or LH-HS2 (DEPC, 3% w/v); the flux values of EFN penetrated for 20 h after the application of LH-HS1, LH-HS2, and LH-HS3 were estimated to be 115.0, 81.8, and 161.0 μg∙cm-2·h-1, respectively (Table 4). The permeation of EFN through the hoof slice was further increased as the proportion of HP-β-CD increased (6% w/v, LH-HS4), providing a flux value of 200.2 μg∙cm-2·h-1.” As mentioned in the previous comment, this statement should be supported by a statistical analysis in which the mentioned 4 means are compared, and that comparison should appear in the manuscript.
6. Lines 159 – 166: “(…) the amount of EFN permeation after topical application of L-HSs was over 41% (L-HS3), 33% (L-HS2), 24% (L-HS4), and 13% (L-HS5) higher than that of the HS base after 20 h, respectively (Table 4). In particular, L-HS3 containing Labrafac PG as a lipophilic enhancer exhibited the highest permeability profile over the experimental period, showing 36% higher permeation than the commercialized composition (L-HS1) after 20 h. Labrafac PG is an oily ingredient with hydrophilic-lipophilic balance (HLB) value of 2.0, which has been employed as an oil vehicle in topical formulations as an effective solubilizer.” As the statistical analysis did not find differences with statistical significance when comparing each L-HS1 to L-HS6 formulations with the HS base formulation (Table 4, in which there are no asterisks in the L-HS formulations), the authors cannot state that the values for permeated EFN (or for flux) of the L-HS formulations are higher than those of the HS base formulation.
NEW COMMENTS
– Lines 83-86: Up to this point of the Introduction, onychomycosis is presented in relation to toenails. Then, in this paragraph, for the first time in the Introduction, the authors change from the toenail to the bovine hoof without any explanation: “The objective of the present study was to design a topical preparation for EFN with high ungual permeation profile. Different hydroalcoholic solutions (HSs) containing several lipophilic enhancers (L-HSs, Table 1) were formulated, and their physicochemical properties and permeation profile through the bovine hooves were evaluated. After selecting a lipophilic enhancer (…)". In the Introduction, it should be made clear that the bovine hoof is employed as a model of the human toenail.
– Lines 86 – 90: “After selecting a lipophilic enhancer, HSs containing both lipophilic and hydrophilic permeation enhancers (LH-HSs, Table 2) were further designed, and their permeation profiles were evaluated with L-HSs including formula similar to the marketed product (L-HS1), or several oil solutions (OSs, Table 3).” It seems that an “a” is missing before “formula”.
– Lines 155 – 159: The authors state “The overall amount of EFN permeated was higher in L-HS formulations, including L-HS3 (Labrafac PG, flux value of 138.1 μg∙cm-2·h-1), L-HS2 (Lauroglycol 90, 129.9 μg∙cm-2·h-1), L-HS4 (Medium-chain triglyceride, 120.8 μg∙cm-2·h-1), and L-HS1 (marketed formula, 112.0 μg∙cm-2·h-1), compared to HS base (97.6 μg∙cm-2·h-1); (…)”. This sentence is confusing, since it seems that the authors are employing flux as a measure of the “overall amount of EFN permeated”, which is not correct. Moreover, if the authors intend to compare the mass of EFN permeated per square centimeter after 20 h, they need to do a statistical analysis of the results to check if the differences have statistical significance.
– The units of the abs/evap times are missing in all tables in which they appear.
– Table 4: “HL base” should be replaced by “HS base”. Additionally, the authors have employed LH as an abbreviation of lipophilic/hydrophilic but, in several instances in section 2.4, an HL abbreviation does appear. Please check.
– Line 241: OL1–OL3 should be replaced by OS1–OS3.
GENERAL COMMENT CONCERNING THE BEST FORMULATION
The best formulation has an abs/evap time of 70 ± 4 min, and long evaporation times tend to be beneficial for drug penetration. However, such long abs/evap times may be inconvenient, since they could preclude the use of this formulation at home, although they could still be adequate for use in the hospital/clinic. This is because it would imply that nothing could touch the toenail for more than 1 hour, which seems a rather inconveniently large waiting time. In fact, one can find antifungal toenail formulations on sale that just require a 5 min waiting time.
Author Response
Thanks for reviewing the manuscript in detail: Ms. Ref. No.: Molecules. 1060158. The reviewer's comments were carefully studied and reflected in the revised manuscript. We tried our best to follow your precious commentary. Please find the separate sheets attached which addresses, point-by-point, the issues raised by the reviewers.
